# I-DIFF: ISOTROPY-BASED REGULARIZATION FOR GENERATION OF COMPLEX DATA DISTRIBUTIONS

## ABSTRACT

Denoising Diffusion Probabilistic Models (DDPMs) have significantly advanced generative AI, achieving impressive results in high-quality image and data generation. However, enhancing consistency and fidelity remains a key challenge in the field. The conventional DDPM framework solely depends on the $L^2$ norm between additive and predicted noise to generate new data distributions. However, it does not explicitly impose structural information in the data distributions, limiting its ability to capture complex geometries (e.g., multimodality, asymmetry, anisotropy), which are commonly found in generation tasks. To address this limitation, we introduce I-Diff, an improved version of DDPM incorporating a carefully designed regularizer that effectively enables the model to encode structural information and capture the anisotropic nature, preserving the inherent structure of the data distribution. Notably, our method is model-agnostic and can be easily integrated into any DDPM variant. The proposed approach is validated through extensive experiments on DDPM and Latent Diffusion Models across multiple datasets. Empirical results demonstrate a 47% reduction in FID on CIFAR-100 dataset compared to the default DDPM, as well as significant improvements in fidelity (Density and Precision increase 27% and 16% in CIFAR-100 dataset respectively) across other tested datasets. These results highlight the effectiveness of our method in enhancing generative quality by capturing complex geometries in data distributions.

## 1 INTRODUCTION

Diffusion models have been accomplishing great feats in the realm of generative AI, specifically in terms of unconditional and conditional image generation ((Nichol & Dhariwal, 2021; Ho et al., 2022b; Saharia et al., 2022; Nichol et al., 2021; Ramesh et al., 2022; Rombach et al., 2022; Ho & Salimans, 2022; Jeong et al., 2024; Dhariwal & Nichol, 2021)). Starting with the revolutionary paper by Ho et al. (2020) and the improvements by Nichol & Dhariwal (2021) as well as the Latent Diffusion Model by Rombach et al. (2022), these models have had the biggest impact in this context. The fidelity and diversity of the images generated by these models are surprisingly amazing. Yet, as with all models, these models can still be improved upon closer inspection. As with the improvements done by Nichol & Dhariwal (2021) to the original Denoising Diffusion Probabilistic Model (DDPM) by introducing techniques such as the cosine-based variance schedule and allowing the model to learn the variance rather than keeping it fixed, helpes to improve the performance of DDPMs.

Even though DDPMs perform well, we noticed that these existing models do not necessarily incorporate any distributional (structural) information about the particular data distribution it tries to sample from. In general, the DDPM's forward process gradually pushes the dataset towards an isotropic Gaussian, which can be thought of as the structural vanishing point of the data distribution (Nichol & Dhariwal, 2021). This implies a well-placed point of origin for the generative process (reverse path) from a point of complete lack of structure toward the final destination, which is the data distribution. In the DDPM implementation, the learning process considers the expected squared norm ($L^2$) difference between the additive Gaussian noise and the predicted noise as its objective function. Therefore, for the generative process, to enhance the aforementioned creation of structure, the objective function can be modified to include any structural measure, such as isotropy. Our goal in this paper is to make a contribution with regard to the improvement of the important fidelity metrics, Density (Naeem et al., 2020) and Precision (Kynkäänniemi et al., 2019) which are explained in Section 5, by imposing possible regularizations that promote the modified DDPM algorithm to learn the underlying structures, diversity, modality and density spread of the true distribution. Thus, we

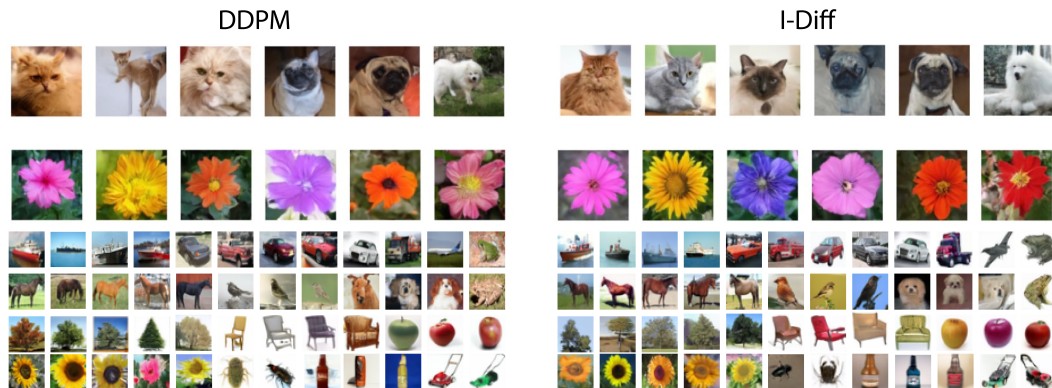

**DDPM**                    **I-Diff**

Figure 1: **Comparison of the generated images via the DDPM (left) and I-Diff (right)**. The DDPM generated images contain much more artifacts and do not seem realistic. However, the generated images via I-Diff are much more realistic and thus, they are of high fidelity.

were motivated to include the isotropic nature of the additive Gaussian noise when optimizing for the objective to further enhance the statistical properties of the predicted noise through backpropagation. The current objective function of the DDPM does not include any mechanism that explicitly encourages the isotropic nature of the predicted noise. Therefore, a mechanism that guarantees the model progresses from a more anisotropic distribution (distributions with multiple modes, non-uniformly distributed spatial structures) to an isotropic Gaussian distribution toward the vanishing point in a structured and learned manner is needed. Our intuition is that by capturing the statistical properties of the noise in more detail, the model will be able to produce higher-fidelity samples as it would have much more information regarding the distributional structure of the samples.

As the rationale for introducing isotropy to the objective function has been established, now, let us see how isotropy establishes convergence and quantifies structural information about the distribution. For example, the isotropy of an isotropic random vector in $\mathbb{R}^n$ is the expected squared norm of that vector, which is equal to its dimension, $n$ (Vershynin, 2018). This demonstrates the convergence towards a normalized distribution with a complete lack of structure (i.e, isotropic). Conversely, the desired distribution, which has more structure and is more antropic, would consequently have a lower isotropy value. This implies that the generative process, in its drive towards a structural distribution, minimizes isotropy.

The inclusion of this constraint does not incur a large computational cost and can be readily applied to any diffusion model variant. In this work, we scrutinize the behavioral aspects of the DDPM model to interpret its functionality using well-defined 2D synthetic datasets, such as Swiss Roll, Scattered Moon, Moon with Two Circles, and Central Banana, drawing fundamental conclusions about the DDPM algorithm. Additionally, we experiment with our modified objective function on these four synthetic datasets and extend our validation to unconditional image generation.

The contributions of this work are as follows:
- We introduce I-Diff: a modified approach that introduces an isotropic constraint on the predicted noise objective function to steer the generative process in a structurally coherent manner. This results in improved fidelity of the generated data distribution. We believe, to the best of our knowledge, that we are the first to propose such a modified loss based on the structural properties of the noise.

- We analyze the simple loss function in the DDPM and its connection to isotropy. Moreover, we show that the isotropy of the data distribution monotonically increases and converges to the maximum isotropy value, which corresponds to an isotropic Gaussian distribution. This confirms that the definition of isotropy mentioned in this paper, conveys information about the structure of the data distribution when the data distribution undergoes the forward process in DDPMs.

- We evaluate and validate our approach on four 2D synthetic datasets as well as on the task of unconditional image generation on Oxford Flower (Nilsback & Zisserman, 2008), Oxford-IIIT Pet (Parkhi et al., 2012), CIFAR-10 (Krizhevsky et al.) and CIFAR-100 (Krizhevsky et al.) datasets. Considering the key evaluation metrics, such as Precision and Recall

(Kynkäänniemi et al., 2019), Density and Coverage (Naeem et al., 2020), Frechet Inception Distance (FID) (Heusel et al., 2017) and Inception Score (IS) (Salimans et al., 2016), the modified objective is able to surpass the original DDPM with a significant gap in terms of the fidelity metrics, Density and Precision.

- We conduct an in-depth analysis of the Density and Coverage metrics to evaluate the generative capabilities of I-Diff compared to DDPM. This analysis facilitates a detailed comparison between the generated and true data distributions, visually illustrating I-Diff's superior alignment with the true distribution. Furthermore, it highlights the importance of these metrics for assessing generative AI algorithms in computer vision applications.

## 2 RELATED WORK

Generative models, particularly in recent years, have gained significant momentum due to their applications in various fields. They began with specific use cases and have evolved along a clear trajectory, as outlined below.

**Deep Generative Models** like GANs (Goodfellow et al., 2020), VAEs (Kingma & Welling, 2013), flow-based models (Rezende & Mohamed, 2015), autoregressive models (Salimans et al., 2017) and diffusion models (Ho et al., 2020; Sohl-Dickstein et al., 2015) learn the probability distribution of given data, allowing us to sample new data points from the distribution. Deep generative models have been used for generating images, videos (Ho et al., 2022a), 3D objects (Mo et al., 2023), audio and speech (Kong et al., 2021), crystal structures (Yang et al., 2024), etc. Moreover, these models have been used for inverse problem solving (Song et al., 2021; Laroche et al., 2023) and to understanding the latent representations of the distributions.

**Diffusion Models**, in particular, have been making huge improvements and have been used in many domains due to their high generative capabilities. There are mainly two types of diffusion models, one is the Score based approach introduced by Song & Ermon (2019) and the other, which is the focus of this work, is the one introduced by Ho et al. (2020). Both modeling types have been able to achieve state-of-the-art performance in generative modeling tasks and have motivated the growth of many subsequent works in generative models.

**Improving Diffusion Models:** In the context of DDPMs (Ho et al., 2020), there have been several works contributed to advance DDPMs beyond their original formulation. Improved diffusion models (Nichol & Dhariwal, 2021) introduced techniques such as cosine-based variance schedules to enhance training stability and sample quality. Variational diffusion models further extended this direction by proposing a unifying framework with refined objectives. Denoising Diffusion Implicit Models (DDIM) (Song et al., 2022) provided a non-Markovian formulation that allows deterministic sampling, while Pro-DDPM (Salimans & Ho, 2022) accelerated generation through progressive distillation. Complementing these methods, the Min-SNR weighting strategy (Hang et al., 2023) improved training efficiency by adapting the importance assigned to different noise levels.

More recent research has focused on modifying the noise design in the forward diffusion process. Blue-noise diffusion (Huang et al., 2024) replaced standard Gaussian noise with structured blue-noise patterns to produce perceptually sharper generations. Edge-preserving diffusion models (Vandersanden et al., 2025) introduced noise schemes that emphasize boundaries and fine details. Extending these ideas to non-Euclidean domains, directional diffusion models (Yang et al., 2023) applied noise injection strategies tailored to graph structures, enabling more expressive representation learning and recommendation (Yi et al., 2024).These approaches highlight that ongoing research continues to seek further improvements at the fundamental level of diffusion design.

However, most of these improvements were focused on improving the models based on the most widely used metrics for image generation, FID and IS. But some of the recent work ((Kynkäänniemi et al., 2019; Naeem et al., 2020; Rosasco et al., 2024)), in generative models has pointed out that FID and IS are not necessarily indicative of the actual fidelity of the samples generated by generative models. Thus, researchers have been focusing on finding other metrics, such as Precision and Density, to assess the fidelity of these generated samples (Dufour et al., 2024; Singh et al., 2024). In particular, we observed that the Density takes the local context (measuring how close it is to densely packed samples of the true distribution) of a sample into account during its calculation. We believe that this makes the Density a vital metric to assess the samples' fidelity.

## 3 BACKGROUND

### 3.1 DEFINITIONS

In the DDPM, we simply add a Gaussian noise, which varies according to a specific variance schedule $\beta_t \in (0,1)$. The noise at each time step corrupts the data, such that by the time the time step reaches its final value $T$, the data will be mapped to an almost isotropic Gaussian distribution. However, the learning occurs when we try to learn the reverse process by which we try to denoise along the same trajectory starting from the almost isotropic Gaussian distribution. The first process, in which we add noise, is called the forward process and the latter, in which we denoise, is called the reverse process. The forward process is often characterized by $q$ and the reverse process by $p$. Both of which are modeled as Gaussian distributions.

The forward process is defined as follows,

$$q(x_1, x_2, \ldots x_T | x_0) = \prod_{t=1}^{T} q(x_t | x_{t-1}) \tag{1}$$

$$q(x_t | x_{t-1}) \sim \mathcal{N}(x_t; \sqrt{1 - \beta_t} x_{t-1}, \beta_t \mathbf{I}) \tag{2}$$

Moreoever, by introducing $\alpha_t = 1 - \beta_t$ as well as $\bar{\alpha}_t = \prod_{i=1}^{t} \alpha_i$ the forward process can be further simplified into the following expression via the re-parametrization trick (Kingma & Welling, 2013). Since,

$$q(x_t | x_{t-1}) \sim \mathcal{N}(x_t; \sqrt{1 - \beta_t} x_{t-1}, \beta_t \mathbf{I}) \tag{3}$$

$$q(x_t | x_0) \sim \mathcal{N}(x_t; \sqrt{\bar{\alpha}_t} x_0, \sqrt{1 - \bar{\alpha}_t} \mathbf{I}) \tag{4}$$

$$x_t = \sqrt{\bar{\alpha}_t} x_0 + \sqrt{1 - \bar{\alpha}_t} \epsilon \tag{5}$$

where, $\epsilon \in \mathcal{N}(0, \mathbf{I})$.

The reverse process, given by $p \sim \mathcal{N}(x_{t-1} | x_t)$, can be obtained in terms of the forward process distribution $q$ and Baye's Theorem. However, the reverse process only becomes tractable when the posterior distribution $q(x_{t-1} | x_t)$, is conditioned on the input data $x_0$. Thus, during training, the model tries to learn the tractable $q(x_{t-1} | x_t, x_0)$ distribution. This distribution, which is also a Gaussian distribution, is defined by the following equation and parameters.

$$q(x_{t-1} | x_t, x_0) \sim \mathcal{N}(x_{t-1}; \tilde{\mu}(x_t, x_0), \tilde{\beta}_t \mathbf{I}) \tag{6}$$

$$\tilde{\beta}_t = \frac{1 - \bar{\alpha}_{t-1}}{1 - \bar{\alpha}_t} \beta_t \tag{7}$$

$$\tilde{\mu}_t(x_t, x_0) = \frac{\sqrt{\bar{\alpha}_{t-1}} \beta_t}{1 - \bar{\alpha}_t} x_0 + \frac{\sqrt{\alpha_t}(1 - \bar{\alpha}_{t-1})}{1 - \bar{\alpha}_t} x_t \tag{8}$$

### 3.2 TRAINING PROCESS

To train, however, one could make the model predict the mean of the reverse process distribution at each time step. But Ho et al. (2020) mentions that predicting the additive noise, $\epsilon$, leads to better results. The additive noise and the mean of the reverse process distribution at each time step are elegantly linked by Equations 5 and 8. This results in the following re-parametrization of $\tilde{\mu}(x_t, t)$,

$$\tilde{\mu}(x_t, t) = \frac{1}{\sqrt{\alpha_t}} \left( x_t - \frac{1 - \alpha_t}{\sqrt{1 - \bar{\alpha}_t}} \epsilon \right) \tag{9}$$

Therefore, predicting the additive noise $\epsilon$, is adequate for the task of predicting the mean of the backward process distribution. Moreover, since the forward process' variance schedule is fixed, the reverse process variance, $\tilde{\beta}_t$, is also assumed to be fixed according to $\tilde{\beta}_t$.

Thus, Ho et al. (2020) proposes to optimize the following simple objective function during the training process.

$$L_{\text{simple}} = \mathbb{E}_{t, x_0, \epsilon}[||\epsilon - \epsilon_\theta(x_t, t)||^2] \tag{10}$$

where $\epsilon_\theta(x_t, t)$ is the predicted noise.

### 3.3 WHY ISOTROPIC STRUCTURAL REGULARIZER

Let us reformulate the diffusion framework on a metric space to reimagine its underlying mechanisms visually. Consider a metric space where each point represents a distribution, and the distance between points is measured using the $L^2$ norm. Furthermore, we assume that the origin of this space corresponds to an isotropic Gaussian distribution ($p(\cdot)$), which reflects the structural vanishing point of the forward diffusion process. Mathematically, for a given predicted noise distribution, the distance from the origin is

$$d_{L^2}(p_{\theta,t}(\cdot), p(\cdot), t) = \left( \int (p_{\theta,t}(y) - p(y))^2 \, dy \right)^{1/2} \tag{11}$$

Here, $p_{\theta,t}(\cdot)$ is the probability distribution function of the predicted noise ($\epsilon_\theta$) for a given data point $x_t$ at time sample $t$. It is obtained as the solution to the minimization problem defined in Equation 10. This distance quantifies how far a given distribution is from an isotropic Gaussian distribution. The reverse diffusion process can thus be seen as a path in this space, starting at origin, and moving toward the desired distribution, with the $L^2$ norm providing a metric structure. For this to be a metric space, the distance must satisfy certain conditions mentioned and derived in the Appendix A.1. The original DDPM algorithm, at the $k^{th}$ step of the reverse process, predicts a noise distribution by learning how to map adjacent points in a metric space that represents the difference between $x_k$ and $x_{k-1}$ (as in Equation 10). However, this formulation is limited to learning only the $L^2$ norm disparity. Consequently, if two distributions share the same $L^2$ norm but differ in an anisotropic manner within this metric space, the algorithm remains agnostic to those differences (See Figure 2(a)). Building upon this, if the transition from $x_k$ to $x_{k-1}$ is defined in terms of an $L^2$ norm, multiple anisotropic distributions with the same $L^2$ norm may exist at the same point within this space. As a result, an $L^2$-norm-based metric space alone may fail to distinguish structural differences between the generated data distributions. Under this formulation, the reverse process iteratively refines $x_k$ to $x_{k-1}$ until the desired final distribution, $x_0$ is generated, ultimately forming the final data distribution, which might be handicapped by the aforementioned limitations. To address this, incorporating an

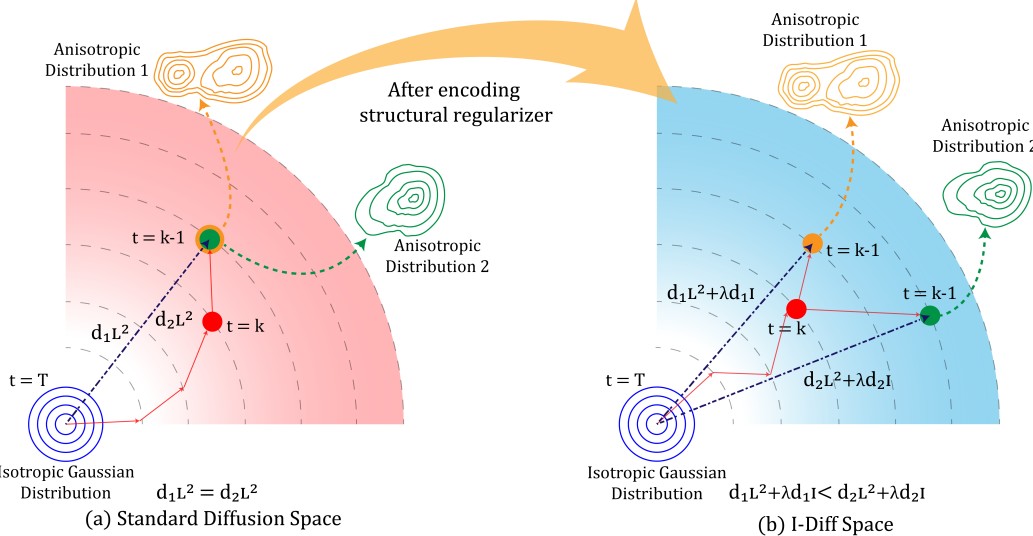

(a) Standard Diffusion Space  (b) I-Diff Space

Figure 2: **How the isotropy-based regularizer guides the model toward a more structurally coherent data distribution compared to the reverse process of the standard DDPM**: The left side of the figure illustrates that during the reverse diffusion process, when the model encounters equidistant distributions at a particular time step, it is not explicitly guided to select the most structurally coherent distribution, leading to a mismatch with the true underlying data distribution. On the right side, when the isotropy constraint (I-Diff) is applied, previously equidistant distributions are now separated by distinct distances. This explicit structural enforcement enables the reverse diffusion process to steer towards a distribution that more accurately aligns with the true data distribution.

isotropic measure into the metric space allows the model to account for both $L^2$ norm disparity and isotropic variations. This enables the predicted noise to encode structural differences more effectively. Mathematically, the distance in the metric space after encoding the structural regularizer (I-Diff Space) is,

$$d_{\text{new}}(p_{\theta,t}(\cdot), p(\cdot), t) = d_{L^2}(p_{\theta,t}(\cdot), p(\cdot), t) + \lambda \cdot d_I(p_{\theta,t}(\cdot), p(\cdot), t) \tag{12}$$

Here, $d_{L^2}(p_{\theta,t}(\cdot), p(\cdot), t)$ denotes the default $L^2$ norm distance, and $d_I(p_{\theta,t}(\cdot), p(\cdot), t)$ denotes the isotropic measure that captures the disparity in the predicted noise for a given $x_t$ at time sample $t$ compared to isotropic Gaussian distribution $p(\epsilon)$, and $\lambda$ is the regularization parameter. It can be derived that I-Diff space becomes a metric space as mentioned in the Appendix A.1. As a result, after noise adjustment, the generated data distributions reflect not only $L^2$ norm-based deviations but also structural disparities. In this revised metric space, two distributions with identical $L^2$ norms but distinct isotropic properties can be mapped to separate points, as illustrated in Figure 2(b), thereby facilitating enhanced structural encoding on the generated data distribution. In this structurally enriched metric space, the final distribution, $x_0$ which is recursively formed by transitioning from $x_k$ to $x_{k-1}$ while imposing isotropic properties of the distribution. This process inherently provides a greater capability to capture the complex geometries of the data distribution, which is reflected in the results, even with a computationally inexpensive amendment to the objective function.

## 4 ISOTROPY BASED LOSS FUNCTION

In the default DDPM model, the variance schedule drives the transformation toward an isotropic Gaussian distribution by restricting the degrees of freedom for the movement of information of the distribution, without using backpropagation to adaptively learn the degree of isotropy achieved, making it, a non-learnable process. With the identified need for an isotropic structural regularizer, Vershynin (2018) suggested the expected squared norm of $\epsilon$ 13 as an isotropic measure.

$$Isotropy = \mathbb{E}(\epsilon^T \epsilon) \tag{13}$$

Then, we proceeded to modify the objective function $L_{\text{simple}}$ to include a regularization term which penalizes the model, if the model predicts a noise which is not necessarily isotropic. Hence, the new modified objective function we propose to optimize is,

$$L_{\text{modified}} = \mathbb{E}(||\epsilon - \epsilon_\theta||^2) + \lambda(\mathbb{E}(\epsilon_\theta^T \epsilon_\theta) - n)^2 \tag{14}$$

where $\lambda$ is the regularization parameter. Under the assumption of a complex optimizer surface with many local minima, the additional regularizer helps guide the optimizer toward more effective directions, reducing the likelihood of suboptimal solutions (Goodfellow et al., 2016).

However, this modified objective needs to be further simplified so as to make this new error, independent of the size of the dimension of the random vector. Thus, we make the following modification during implementation.

$$L_{\text{modified}} = \mathbb{E}(||\epsilon - \epsilon_\theta||^2) + \lambda \left( \mathbb{E}\left( \frac{\epsilon_\theta^T \epsilon_\theta}{n} \right) - 1 \right)^2 \tag{15}$$

## 5 INTERPRETATION OF EVALUATION METRICS

Let us define a couple of terms that would be useful to understand the evaluation metrics. A Neighborhood Sphere is the local area around a data point that includes nearby points, defined by a radius that is the distance to the k-th nearest neighbor (see Figure 3). True manifold is created with the collection of neighborhood spheres formed by real data points. Similarly, a generated sample manifold is created with the collection of neighborhood spheres formed by generated data points.

**Precision** denotes the fraction of generated data that lies in the true manifold by counting whether each generated data point falls within a neighborhood sphere of real samples. This measure reflects how closely the generated points align with the true manifold (Kynkäänniemi et al., 2019; Sajjadi et al., 2018). In the left of Figure 3, all four generated points lie within the true manifold; therefore, Precision is 4/4.

**Recall** denotes the fraction of true data that lies in the generated sample manifold. This measure indicates how well the true points align with the generated sample manifold (Kynkäänniemi et al., 2019; Sajjadi et al., 2018). However, the lack of generated samples near sparse outliers in the true data leads to low Recall, as the sample manifold fails to capture these regions. In the right of Figure 3, five out of seven real samples are covered; therefore, Recall is 5/7.

**Density** counts the number of neighborhood spheres of real samples that encompass each generated data point. This allows Density to reward generated samples located in areas densely populated by real samples, reducing sensitivity to outliers. This enables us to consider the local context of a

distribution by measuring how close a sample is to densely packed points in the true distribution (Naeem et al., 2020). In the left of Figure 3, summing the overlaps, which are 3, 2, 2, and 1; therefore, Density equals 8/7.

**Coverage** measures the fraction of real samples whose neighborhoods contain at least one generated sample. Moreover, Coverage measures the diversity of a distribution by assessing whether all aspects of the distribution are represented. However, the presence of sparse outliers in the true manifold and the absence of the generated samples near those outliers may result in low Coverage (Naeem et al., 2020). In the left of Figure 3, five out of seven real samples have a generated sample nearby; therefore, Coverage is 5/7.

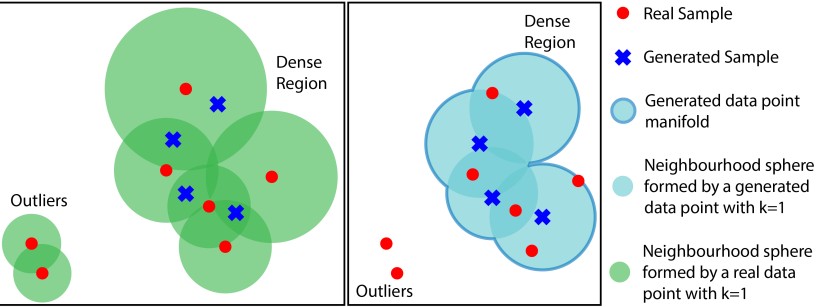

Figure 3: An example scenario for illustrating a situation where high Density and low Coverage is recorded. Generating samples in the neighborhoods of the highly dense regions over the outliers in the true manifold has resulted in a high Density and low Coverage.

## 6 EXPERIMENTS

### 6.1 PERFORMANCE COMPARISON BETWEEN DDPM AND I-DIFF

To compare the performance between DDPM and I-Diff, the modified loss function was utilized in all four 2D synthetic datasets, Oxford Flower dataset, Oxford IIIT Pet dataset and CIFAR-10 dataset. Precision, Recall, Density along with Coverage were used to evaluate and compare the performance of the two models on 2D synthetic datasets. In addition to those four evaluation metrics, FID and IS were used to evaluate the quality of the generated samples for the image datasets Oxford Flower, Oxford-IIIT Pet, CIFAR-10 and CIFAR-100.

Table 1: Comparison of Evaluation Metrics for the two methods: DDPM and I-Diff$_{DDPM}$ for the 2D Datasets

| Metrics | Swiss Roll | | Scattered Moon | | Moon with Two Circles | | Central Banana | |
|---|---|---|---|---|---|---|---|---|
| | DDPM | I-Diff$_{DDPM}$ | DDPM | I-Diff$_{DDPM}$ | DDPM | I-Diff$_{DDPM}$ | DDPM | I-Diff$_{DDPM}$ |
| Precision (↑) | 0.9458 | **0.9893 (+4.60%)** | 0.9990 | **0.9993 (+0.03%)** | 0.9921 | **0.9982 (+0.61%)** | 0.8974 | **0.9072 (+1.09%)** |
| Recall (↑) | **0.9927** | 0.9709 (-2.19%) | **0.9962** | 0.9736 (-2.27%) | **0.9967** | 0.9694 (-2.74%) | **0.9977** | 0.9417 (-5.61%) |
| Density (↑) | 0.8946 | **0.9908 (+10.75%)** | 1.0015 | **1.0049 (+0.34%)** | 0.9925 | **1.0081 (+1.57%)** | 0.8785 | **0.8962 (+2.01%)** |
| Coverage (↑) | **0.8932** | 0.8458 (-5.31%) | **0.9605** | 0.8254 (-14.07%) | **0.9498** | 0.8572 (-9.75%) | **0.9102** | 0.6840 (-24.85%) |

Table 1 demonstrates the comparison between the best performing isotropy based model (I-Diff) and DDPM in terms of the generative model's evaluation metrics along with the percentage change from DDPM. Across all these 2D synthetic datasets we observed that the fidelity metrics, Precision and Density have been improved in I-Diff. The results of Table 2 further confirm the improvements made by our modified loss on the quality of image samples. The Density of the generated images has been significantly improved for all four datasets. Moreover, the FID score has been significantly improved in the CIFAR-10 dataset by the proposed method. Furthermore, the results in Table 3 back up these improvements by showing steady gains with another diffusion model (LDM), proving that our approach works well across different diffusion methods.

Although the performance of the modified loss function has been able to produce samples that surpass the original DDPM's samples quality, the quality depends on the regularization parameter of the modified loss function. In particular, we performed a few more experiments by considering a range of values for the regularization parameter. The metrics for the Oxford Flower dataset and Oxford-IIIT-Pet dataset with different values of the regularization parameter ranging from 0.01 to

0.30 are tabulated in Table 4 and Table 5 in supplementary materials. We observe that, there exists an optimal $\lambda$ that maximizes the overall improvement across these metrics.

Table 2: Comparison of Evaluation Metrics for the two methods: DDPM and I-Diff$_{DDPM}$ for the Image Datasets

| Metrics | Oxford Flower | | Oxford-IIIT-Pet | | CIFAR-10 | | CIFAR-100 | |
|---|---|---|---|---|---|---|---|---|
| | DDPM | I-Diff$_{DDPM}$ | DDPM | I-Diff$_{DDPM}$ | DDPM | I-Diff$_{DDPM}$ | DDPM | I-Diff$_{DDPM}$ |
| FID ($\downarrow$) | 55.590 | **47.310 (-14.9%)** | 34.087 | **31.900 (-6.4%)** | 6.314 | **5.303 (-19.1%)** | 9.619 | **6.543 (-47.0%)** |
| IS ($\uparrow$) | 3.097 | **3.504 (+13.1%)** | 7.083 | **7.531 (+6.3%)** | 8.952 | **8.969 (+0.2%)** | 10.588 | **10.595 (+0.1%)** |
| Precision ($\uparrow$) | 0.725 | **0.944 (+30.3%)** | 0.819 | **0.954 (+16.5%)** | 0.566 | **0.602 (+6.3%)** | 0.502 | **0.585 (+16.4%)** |
| Recall ($\uparrow$) | **0.184** | 0.056 (-69.8%) | **0.152** | 0.063 (-58.4%) | **0.481** | 0.466 (-3.0%) | **0.521** | 0.472 (-9.3%) |
| Density ($\uparrow$) | 2.632 | **11.039 (+319.4%)** | 6.704 | **15.778 (+135.4%)** | 1.214 | **1.313 (+8.2%)** | 0.979 | **1.248 (+27.5%)** |
| Coverage ($\uparrow$) | 0.959 | **0.994 (+3.6%)** | 0.9996 | **0.9999 (+0.03%)** | 0.980 | **0.985 (+0.4%)** | 0.961 | **0.979 (+1.9%)** |

Table 3: Comparison of Evaluation Metrics for the two methods: LDM and I-Diff$_{LDM}$ for the Image Datasets

| Metrics | CIFAR-10 | | CIFAR-100 | |
|---|---|---|---|---|
| | LDM | I-Diff$_{LDM}$ | LDM | I-Diff$_{LDM}$ |
| FID ($\downarrow$) | 10.447 | **10.361 (-0.8%)** | 15.597 | **14.591 (-6.4%)** |
| IS ($\uparrow$) | 8.602 | **8.610 (+0.1%)** | 9.314 | **9.424 (+1.2%)** |
| Precision ($\uparrow$) | 0.674 | **0.712 (+5.6%)** | 0.662 | **0.680 (+2.8%)** |
| Recall ($\uparrow$) | **0.396** | 0.370 (-6.4%) | **0.399** | 0.380 (-4.7%) |
| Density ($\uparrow$) | 1.726 | **2.013 (+16.6%)** | 1.729 | **1.900 (+9.9%)** |
| Coverage ($\uparrow$) | 0.992 | **0.995 (+0.3%)** | 0.989 | **0.992 (+0.3%)** |

Although the FID and IS are considered to be the most widely used evaluation metrics for assessing image generation, we see that in the case of all four datasets, they convey little to no discerning information about the generative ability of the proposed method and the original DDPM. But, by using other metrics such as the Precision, Recall, Density and Coverage (PRDC), we can state that while our proposed method suffers a bit in terms of Recall, the generated samples, are very close to being real (see Figure 1), as indicated by the improvements in the Precision and Density metrics.

## 6.2 INTERPRETATION OF THE RESULTS OF 2D DATA DISTRIBUTIONS USING PRDC VALUES

We believe that the disparity in the changes of Precision, Recall, Density and Coverage is a direct consequence of imposing a structural constraint on the objective function. It is evident that by focusing on the structure or the isotropy of the distribution, our method is capable of capturing highly dense mode regions and generating samples near them rather than being too diverse. Thus, it increases the fidelity but decreases the diversity of the generated samples.

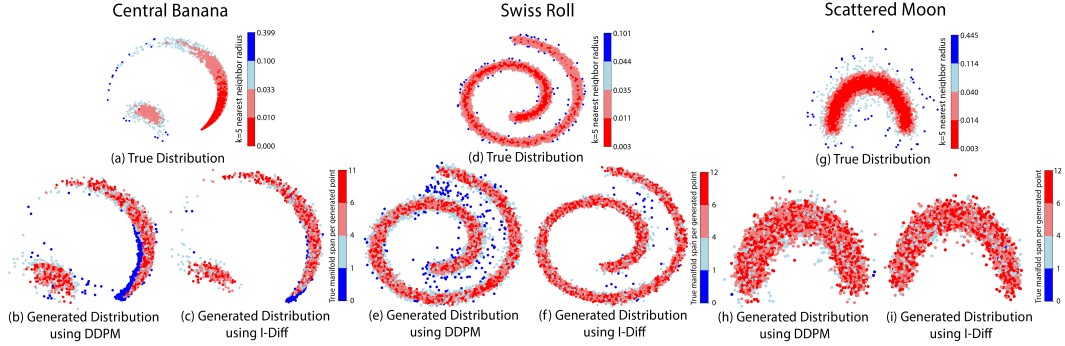

Figure 4: **Central Banana, Swiss Roll, and Scattered Moon 2D synthetic datasets:** (a),(d),(g) True distribution points, color-coded by k=5 nearest neighbor radius. (b),(e),(h) DDPM-generated points, color-coded by true manifold span per point. (c),(f),(i) I-Diff generated points, color-coded by true manifold span per point.

As illustrated in the Figure 4(a), the Central Banana distribution was designed by introducing a distinct mode to the main structure of the distribution resulting in a multimodal distribution with

a density gradient. Once, it is generated via I-Diff as indicated in Figure 4(c), it is evident that, I-Diff, is capable of capturing the main structure even with the discontinuities of the density gradient. However, the illustrations show that DDPM lacks the capability of capturing the discontinuity in the density gradient between the tail end of the main distribution and the distinct mode. Instead, it tries to generate data points that are actually not even in the true distribution by interpolating along the main lobe's trend (see Figure 4(b)). Moreover, the limited capability to capture the discontinuity in the density gradient of DDPM can be further observed in the Swiss Roll distribution 4(d) as well (see Figure 4(e) and 4(f)). The increase in Density and decrease in Coverage for the datasets Swiss Roll and Central Banana are clear evidence for the aforementioned observations. Hence, it is limited in ability to capture the underlying structure of the distribution. Additionally, there is a noticeable trend of generating data points (blue points in Figure 4(b), 4(c)), outside the boundaries of the highly dense regions of the main lobe. This effect is likely due to the model's focus on these high-density regions. However, compared to DDPM, I-Diff effectively regulates the overgeneration of data points outside the boundaries of densely packed regions. This improvement is likely a result of the added regularization in the improved object function, which encourages capturing the main semantics of the true distribution.

As illustrated in the Figure 4(g), the Scattered Moon distribution was designed by imposing scattered noise around the main structure of the data distribution. Once, it is generated via I-Diff as indicated in the Figure 4(i), it is evident that, the model has tried to only capture the underlying semantics of the distribution without being susceptible to the low probable regions. Whereas, the DDPM model shows limitations in capturing the distinction between the main structure and the scattered noise (see Figure 4(h)). The increased Density and reduced Coverage values support this observation. This shows that the proposed objective function, enforces the generated samples to contain properties that push them to be closely linked to the real data. Thus, we can directly observe an improvement in the Density metric as it measures the sample fidelity. We believe that in the context of unconditional image generation, the isotropy based objective function helps the model learn to keep the generated samples closer to the high-density regions of the ground-truth distribution.

These observations highlight the proposed algorithm's ability to increase Density by focusing on the dense regions of the true distribution. At the same time, the absence of generated data in the neighborhoods of low probable data points in the true distribution may result in a reduction in Coverage. When scattering is minimal, Coverage remains consistent. This indicates that the algorithm effectively captures the main structure of the true distribution without extending into low probable regions. Also, each of these metrics has their own utility depending on the application (Kynkäänniemi et al., 2019). Thus, this should motivate the research community to propose new evaluation metrics such as Density, which is a much more meaningful measure of fidelity over FID and IS, to assess generative models.

Preserving the modality of a data distribution is essential, as failing to capture it can lead to a loss of semantic details or edge information, both of which represent high-level features in computer vision and image processing tasks (Guo et al., 2020; Shaham et al., 2019)

## 7 CONCLUSION

Denoising Diffusion Probabilistic Models (DDPMs) excel in generative tasks like image reconstruction and inverse problem solving. However, evaluating Generative AI demands more than just FID metrics like Precision, Recall, Density, and Coverage reveal shortcomings in most diffusion models. Notably, DDPMs struggle to impose structural constraints on the reverse process, limiting their ability to capture complex data geometries. To address this, we propose the I-Diff regularizer, which enhances the ability to capture complex geometries effectively. This aligns the generated data distribution more closely with the true distribution. Testing on DDPM and LDM shows improved FID, IS, Precision, and Density demonstrating that our modified loss function is framework-independent and better captures data fidelity. We firmly believe that our work provides other researchers with novel perspectives into the workings of DDPMs and highlights the effectiveness of PRDC measures in evaluating generative models, paving the way for more robust diffusion-based approaches.

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

## A   APPENDIX

### A.1   VERIFYING THE STANDARD DIFFUSION SPACE AND I-DIFF SPACE AS METRIC SPACES

$$d_{L^2}(p, q) = \left[ \int (p(y) - q(y))^2 \, dy \right]^{\frac{1}{2}} \tag{16}$$

As mentioned in Section 3.3, for the standard diffusion space, defined by the distance $d_{L^2}(p, q)$ to qualify as a metric space, the distance must satisfy the following properties:

**Non-negativity**

$$(p(y) - q(y))^2 \geq 0 \quad ; \text{ as } p(y), q(y) \in \mathbb{R}$$

$$\int (p(y) - q(y))^2 \, dy \geq 0 \quad \text{(integrating a non-negative quantity cannot yield a negative result)}$$

$$\left[ \int (p(y) - q(y))^2 \, dy \right]^{\frac{1}{2}} \geq 0$$

$$\therefore d_{L^2}(p, q) \geq 0 \tag{17}$$

$d_{L^2}(p, q) = 0$ if and only if $p(y) = q(y)$ for every $y$. Therefore, the $d_{L^2}$ distance is zero only when the functions are essentially identical.

Hence, $d_{L^2}(p, q)$ satisfies the non-negativity property.

**Symmetry**

$$d_{L^2}(p, q) = \left[ \int (p(y) - q(y))^2 \, dy \right]^{1/2}$$

$$= \left[ \int (-(q(y) - p(y)))^2 \, dy \right]^{1/2}$$

$$= \left[ \int (q(y) - p(y))^2 \, dx \right]^{1/2}$$

$$= d_{L^2}(q, p) \tag{18}$$

$$\therefore \quad d_{L^2}(p, q) = d_{L^2}(q, p) \tag{19}$$

Hence, $d_{L^2}(p, q)$ satisfies the symmetry property.

**Triangle inequality**

$$d_{L^2}(p, r) = \left[ \int (p(y) - r(y))^2 \, dy \right]^{\frac{1}{2}}$$

$$d_{L^2}^2(p, r) = \int (p(y) - r(y))^2 \, dy$$

$$= \int ((p(y) - q(y)) + (q(y) - r(y)))^2 \, dy$$

$$= \int (p(y) - q(y))^2 dy + \int (q(y) - r(y))^2 dy$$

$$+ 2 \int (p(y) - q(y))(q(y) - r(y)) \, dy \tag{20}$$

However, by the Cauchy–Schwarz inequality:

$$\left| \int (p(y) - q(y))(q(y) - r(y)) \, dy \right| \leq \sqrt{\int (p(y) - q(y))^2 \, dy} \cdot \sqrt{\int (q(y) - r(y))^2 \, dy}$$

$$= d_{L^2}(p, q) \cdot d_{L^2}(q, r) \tag{21}$$

By applying the result in Equation equation 21 to Equation equation 20, we obtain;

$$2 \int (p(y) - q(y))(q(y) - r(y)) \, dy \leq 2d_{L^2}(p,q) \cdot d_{L^2}(q,r) \tag{22}$$

$$\therefore \quad d_{L^2}^2(p,r) \leq d_{L^2}^2(p,q) + d_{L^2}^2(q,r) + 2d_{L^2}(p,q) \cdot d_{L^2}(q,r)$$

$$= [d_{L^2}(p,q) + d_{L^2}(q,r)]^2$$

Hence,

$$d_{L^2}(p,r) \leq d_{L^2}(p,q) + d_{L^2}(q,r) \tag{23}$$

i.e. $d_{L^2}(p,q)$ satifies the triangle inequality condition. Thus, $d_{L^2}(p,q)$ qualifies as a metric space.

Now, let us define the new distance measure with isotropy regularizer as:

$$d_{\text{new}}(p,q) = d_{L^2}(p,q) + \lambda \cdot d_I(p,q) \tag{24}$$

Where:

$$d_{L^2}(p,q) = \left[ \int (p(y) - q(y))^2 \, dy \right]^{\frac{1}{2}} \quad and \quad d_I(p,q) = |I(p) - I(q)|$$

Here, $I$ is the isotropic measure mentioned in the Section 4, and $\lambda(> 0)$ is the regularization parameter.

As mentioned in Section 3.3, for the new distance $d_{\text{new}}(p,q)$ in the I-Diff Space, to qualify as a metric space, the distance must satisfy the following properties:

**Non-negativity**
According to the result in Equation equation 17, $d_{L^2}(p,q) \geq 0$.
Since $I(p), I(q) \in \mathbb{R}$, we have $|I(p) - I(q)| \geq 0$. $\therefore d_I(p,q) \geq 0$.
Thus,

$$d_{\text{new}}(p,q) \geq 0$$

Hence, $d_{\text{new}}(p,q)$ satisfies the non-negativity property.

**Symmetry**
According to Equation equation 19, $d_{L^2}(p,q) = d_{L^2}(q,p)$.
Also, $|I(p) - I(q)| = |I(q) - I(p)|$. $\therefore d_I(p,q) = d_I(q,p)$.
Thus,

$$d_{\text{new}}(p,q) = d_{\text{new}}(q,p)$$

Hence, $d_{\text{new}}(p,q)$ satisfies the symmetry property.

**Triangle inequality**
According to the Equation equation 23,

$$d_{L^2}(p,r) \leq d_{L^2}(p,q) + d_{L^2}(q,r) \tag{25}$$

Since $I(p), I(q), I(r) \in \mathbb{R}$, we have

$$|I(p) - I(r)| = |(I(p) - I(q)) + (I(q) - I(r))| \leq |I(p) - I(q)| + |I(q) - I(r)| \tag{26}$$

From Equations equation 25 and equation 26, it follows that:

$$d_{L^2}(p,r) + \lambda \cdot d_I(p,r) \leq d_{L^2}(p,q) + \lambda \cdot d_I(p,q) + d_{L^2}(q,r) + \lambda \cdot d_I(q,r)$$

$$\therefore \quad d_{\text{new}}(p,r) \leq d_{\text{new}}(p,q) + d_{\text{new}}(q,r)$$

Hence, $d_{\text{new}}(p,q)$ satisfies the triangle inequality.

Thus, both the standard diffusion space with the $d_{L^2}(p,q)$ distance and the new I-Diff space with the $d_{\text{new}}(p,q)$ distance measure are metric spaces.

## A.2 ADDITIONAL RESULTS

Table 4: Metrics Variation with the Regularization Parameter for the Oxford Flower Dataset ($\lambda$)

| Method | FID ($\downarrow$) | IS ($\uparrow$) | Precision ($\uparrow$) | Recall ($\uparrow$) | Density ($\uparrow$) | Coverage ($\uparrow$) |
|---|---|---|---|---|---|---|
| DDPM | 55.590 | 3.097 | 0.725 | 0.184 | 2.632 | 0.959 |
| I-Diff $\lambda = 0.01$ | 53.337 | 3.202 | 0.784 | 0.157 | 3.345 | 0.976 |
| I-Diff $\lambda = 0.05$ | 54.706 | 3.221 | 0.733 | 0.176 | 2.638 | 0.959 |
| I-Diff $\lambda = 0.10$ | **47.310** | **3.504** | 0.944 | 0.056 | 11.039 | 0.994 |
| I-Diff $\lambda = 0.30$ | 51.582 | 3.311 | **0.946** | **0.055** | **12.544** | **0.995** |

Table 5: Metrics Variation with the Regularization Parameter for the Oxford-IIIT-Pet Dataset ($\lambda$)

| Method | FID ($\downarrow$) | IS ($\uparrow$) | Precision ($\uparrow$) | Recall ($\uparrow$) | Density ($\uparrow$) | Coverage ($\uparrow$) |
|---|---|---|---|---|---|---|
| DDPM | 34.087 | 7.083 | 0.819 | **0.152** | 6.704 | **0.9996** |
| I-Diff $\lambda = 0.01$ | 32.728 | **7.530** | 0.881 | 0.123 | 7.974 | 0.9991 |
| I-Diff $\lambda = 0.05$ | 32.488 | 7.516 | 0.863 | 0.126 | 8.035 | 0.9997 |
| I-Diff $\lambda = 0.10$ | 33.341 | 7.481 | 0.910 | 0.102 | 10.635 | 1.0000 |
| I-Diff $\lambda = 0.30$ | **31.900** | 7.531 | **0.954** | 0.063 | **15.778** | 0.9999 |

## A.3 IMPLEMENTATION DETAILS

### A.3.1 EXPERIMENTAL SETUP

To validate our approach, we consider 2D synthetic data as well as images. For the 2D data, we utilized a conditional dense network consisting of 3 fully-connected hidden layers with ReLU activations instead of the conventional U-Net model as the noise prediction model. The learning rate was fixed at $1 \times 10^{-3}$. All the datasets were learned using 1000 time-steps and 1000 epochs.

Evaluation metrics are reported as the average of 3 training runs per dataset, with Precision, Recall, Density and Coverage (PRDC) values calculated using k=5 nearest neighbors for each dataset. Moreover, all the experiments were run on one Quadro GV-100 GPU with 32GB of VRAM.

### A.3.2 IMPLEMENTATION

We implemented diffusion models following the designs of Ho et al. (2020) and Rombach et al. (2022). We used an open-sourced codebase of the Denoising Diffusion Implicit Model for DDPM training and that of Rombach et al. (2021) for latent diffusion training. Our networks are ResNet-based U-Nets with sinusoidal time embeddings. We included self-attention at specified spatial scales and used dropout only where noted below. We maintained an exponential moving average (EMA) of model parameters (decay = 0.9999) for all models. The details for each configuration are as follows.

**DDPM on CIFAR-10 and CIFAR-100**
**Model architecture:** We used a U-Net similar to that of DDPM by Ho et al. (2020). Inputs are $32 \times 32$ RGB images; the network had four spatial resolutions ($32 \times 32 \rightarrow 16 \times 16 \rightarrow 8 \times 8 \rightarrow 4 \times 4$). The base number of channels is 128, which doubles at each downsampling stage. We inserted a self-attention layer at the $16 \times 16$ feature map (the second scale). Dropout ($p = 0.1$) was applied only on CIFAR-10; it was disabled for CIFAR-100.

**Diffusion setup:** We used 1000 timesteps with a linear variance schedule ($\beta_1 = 1 \times 10^{-4}$ to $\beta_T = 2 \times 10^{-2}$). This matches the schedule chosen in Ho et al. (2020). At each step, the U-Net predicts $\epsilon_\theta(x_t, t)$; we trained on the standard variational bound objective and on the objective function mentioned in this.

**Training hyperparameters:** Adam optimizer was used with a learning rate of $2 \times 10^{-4}$ for the optimization. We trained for 790K iterations with a batch size of 128. An EMA of the weights was kept with decay of 0.9999.

**Latent DDPM on CIFAR-10 and CIFAR-100**
**Autoencoder:** We first encoded $32 \times 32$ images into a learned latent space using a convolutional variational autoencoder (AutoencoderKL). The encoder downsamples by 2 to produce a $16 \times 16$

spatial latent with 3 channels (embed_dim = 3). A KL-divergence loss encourages the latent to match a standard normal prior (this is similar to the VAE in Rombach et al. (2022)). The decoder mirrors this structure to reconstruct $32 \times 32$ RGB outputs.

**Latent U-Net:** The diffusion U-Net operates on the $16 \times 16 \times 3$ latent. We set model_channels = 128 (base channels) exactly as in the pixel-space model. We applied self-attention at the full latent resolution ($16 \times 16$). Dropout $p = 0.1$ was again used only for CIFAR-10. The decoder and upsampling mirror the encoder's downsampling with channel multipliers [1, 2, 2, 4].

**Diffusion and Training:** We used the same diffusion settings as above: 1000 timesteps and a linear $\beta$-schedule from $1 \times 10^{-4}$ to $2 \times 10^{-2}$. The optimizer is Adam (learning rate = $2 \times 10^{-4}$), EMA = 0.9999, batch size 128, for 790K iterations. All other training details match the pixel-space runs.

**DDPM on Oxford-IIIT-Pet and Oxford Flower**
**Model architecture:** We used a U-Net architecture similar to Ho et al. (2020). Inputs are $64 \times 64$ RGB images (3 channels). The network operates across four spatial resolutions ($64 \times 64 \rightarrow 32 \times 32 \rightarrow 16 \times 16 \rightarrow 8 \times 8$), with a base channel size of 64 and channel multipliers [1, 2, 4, 8].

**Diffusion setup:** We employed 1000 timesteps with a linear variance schedule ($\beta_1 = 0.0001$ to $\beta_T = 0.02$), following Ho et al. (2020). The U-Net predicts $\epsilon_\theta(x_t, t)$ at each timestep, trained on the standard variational bound objective with a fixed large variance type and on the objective function mentioned in Equation 15.

**Training hyperparameters:** Adam optimizer was used with a learning rate = $2 \times 10^{-4}$ for the optimization. Gradient clipping is applied with a maximum norm of 1.0. Training runs for 230K iterations on the Oxford-IIIT-Pet dataset and 256K iterations on the Oxford Flower dataset, with a batch size of 32. An EMA of model weights is maintained with a decay of 0.999.

Algorithm 1 and 2 show the complete training and sampling procedure with the modified objective function, including the isotropy regularizer.

| **Algorithm 1: Training** | **Algorithm 2: Sampling** |
|---|---|
| 1: **repeat** | 1: $x_T \sim \mathcal{N}(0, \mathrm{I})$ |
| 2: $x_0 \sim q(x_0)$ | 2: **for** $t = T, \ldots, 1$ **do** |
| 3: $t \sim \text{Uniform}(\{1, \ldots, T\})$ | 3: $z \sim \mathcal{N}(0, \mathrm{I})$ if $t > 1$, else $z = 0$ |
| 4: $\epsilon \sim \mathcal{N}(0, \mathrm{I})$ | 4: $x_{t-1} = \frac{1}{\sqrt{\alpha_t}} \left( x_t - \frac{1-\alpha_t}{\sqrt{1-\alpha_t}} \epsilon_0(x_t, t) \right) + \sigma_t z$ |
| 5: Take gradient descent step on $\nabla_\theta \left( \mathbb{E} \|\epsilon - \epsilon_\theta\|^2 + \lambda \left( \mathbb{E} \left( \frac{\epsilon_\theta^T \epsilon_\theta}{n} \right) - 1 \right)^2 \right)$ | 5: **end for** |
| 6: **until** converged | 6: **return** $x_0$ |

## A.4 LIMITATIONS

This is a foundational work, positioned as a seminal contribution to this approach, focusing on introducing the core concept of I-Diff and examining the mathematical and theoretical validity of the proposed isotropic modifier as a proof of concept. The evaluation is conducted on vanilla DDPM and Latent Diffusion models, validated using 2D synthetic datasets and standard real-world image datasets. Our primary focus has been to showcase the ability of this model-agnostic modifier to impose structural information on data distributions, thereby enhancing its capability to capture complex geometries and model intricate data distributions. However, given that such a modifier can be applied in a much more expansive manner, there is significant potential to explore its use across multiple, application-specific datasets, though this is not the main focus of the present study. We encourage future researchers to conduct more application-specific and extensive investigations, whether on more complex architectures or across a wider range of datasets, as an extension of this work. Such efforts would further help assess the feasibility and effectiveness of I-Diff in broader contexts and real-world applications.

## A.5 LLM USAGE

The authors used ChatGPT for language editing and subsequently reviewed and revised the text, taking full responsibility for the final content.

