# OpenReview forum: "I-Diff: Isotropy-Based Regularization for Generation of Complex Data Distributions"
_ICLR.cc/2026/Conference — ICLR 2026 Conference Withdrawn Submission_

### Official Review · Reviewer_FuCW · 2025-10-29

**Soundness:** 1
**Presentation:** 2
**Contribution:** 1
**Rating:** 2
**Confidence:** 5

**Summary:**

The paper proposes **I-Diff**, a DDPM variant that introduces an isotropic penalty in the noise-prediction objective, effectively imposing an isotropic constraint on the learned noise term. The notion of isotropy is illustrated geometrically and its usefulness is motivated heuristically. The proposed approach is empirically evaluated on four two-dimensional synthetic datasets and four image datasets with resolutions of 32×32 or 64×64.

**Strengths:**

1. The forward process is driven by additive isotropic Gaussian noise, and the analytical reverse kernel (which is intractable due to its dependence on the unknown data distribution) is therefore a Gaussian kernel with isotropic covariance. The original DDPM loss does not penalize deviations from isotropy in the learned noise term. Introducing such a penalty is conceptually reasonable and, as demonstrated in the paper, straightforward to implement with negligible computational overhead.

2. The empirical results include a comparison with the baseline DDPM across a reasonable number of datasets and report multiple evaluation metrics beyond FID.

3. The proposed method is clearly described and easy to reproduce based on the information provided in the paper.

**Weaknesses:**

**Weaknesses:**
1. Although the authors claim that the proposed “isotropic constraint” enables the model to capture anisotropic structures in the data, the regularizer actually enforces isotropy of the predicted noise (in a scalar sense). This stands in conceptual tension with the claim of improved anisotropy modeling, since the true reverse diffusion kernels are already isotropic Gaussians, and further enforcing isotropy can only reduce directional variability rather than capture complex data geometries.
2. The employed notion of “isotropy” — defined as a random vector $X$ being isotropic if $\operatorname E[\|X\|^2]=d$ — is extremely weak. Since $\operatorname E[\lVert X\rVert^2]=\lVert\operatorname E[X]\rVert^2+\operatorname{tr}\operatorname{Cov}[X]$, even highly anisotropic distributions satisfy this criterion. For example: (a) $X\sim\mathcal{N}(0,\Sigma)$ with $\Sigma=\operatorname{diag}(1-r,1+r,1,\ldots,1)$ for some $r\ne0$; (b) $X\sim\mathcal{N}(\mu,\Sigma)$ with $0<\lVert\mu\rVert^2<n$ and any covariance matrix $\Sigma$ such that $\operatorname{tr}\Sigma=n-\lVert\mu\rVert^2$; or (c) a Bernoulli variable taking values $\pm\sqrt{n}e_1$ with equal probability — all fulfill the paper’s “isotropy” condition despite being strongly anisotropic.
3. Since the paper aims to capture “structural information in the data distribution” and “complex geometries,” the choice of low-resolution datasets (32×32 and 64×64 images) is questionable, as such images contain barely discernible geometric features. Higher-resolution datasets and perceptually grounded evaluation metrics would have been more suitable for validating a method that claims to improve the modeling of structural information.
4. The briefly mentioned applicability of the same “isotropic constraint” to other diffusion models — particularly LDMs — is dubious. Even if the forward process is isotropic in some sense, the geometry of the latent space is typically highly anisotropic. Enforcing isotropy of the noise term there could be counterproductive. Moreover, the reported metric differences between LDM and “I-Diff-LDM” are sometimes in the second decimal place and therefore statistically insignificant, likely stemming from unconverted numbers rather than a genuine improvement.
5. Section 3.3 is intended to provide a geometric intuition for the “isotropy constraint”, yet it also introduces technical concepts such as special metrics that are defined on the space of densities (with respect to a fixed reference measure) rather than on the space of probability measures, which severely limits their analytical utility. The section also contains additional technical issues and the whole section offers little value to the overall paper. The intended motivation could be integrated into the introduction of Section 4, and the current content of Section 3 (including figures) condensed into two sentences.
6. The paper appears rushed. In addition to numerous typos, there are structural and notational inconsistencies. For example, the sentence “The forward process is often characterized by $q$ and the reverse process by $p$” (line 171) is meaningless, since $q$ and $p$ have not been introduced up to that point. Equation (3) is a duplicate of Equation (2). In Equations (3), (4), and (6), the notation “$\sim$” is incorrectly used instead of “$=$” (which is already deviant from mathematical conventions). In line 187 it should read $\varepsilon\sim\mathcal N(0,I)$. In line 188, an index or argument is missing on the left-hand side of $p\sim\mathcal N(x_{t-1}\mid x_t)$; the symbol $p$ then disappears and only reappears in Section 3.3 — but with a different meaning. This list could easily be extended.

**Questions:**

See weaknesses.

---

### Official Review · Reviewer_zJr9 · 2025-10-30

**Soundness:** 2
**Presentation:** 2
**Contribution:** 2
**Rating:** 2
**Confidence:** 4

**Summary:**

This paper introduces I-Diff, a method that adds a regularization term to the standard DDPM loss function. The regularizer is designed to explicitly enforce that the predicted noise is isotropic by penalizing the deviation of its expected squared norm from the feature dimension. The authors motivate this by arguing that the standard L2 loss does not sufficiently impose structural information, limiting the model's ability to capture complex, anisotropic data distributions. Through experiments on 2D synthetic data and several image datasets (CIFAR-10/100, Flowers, Pets), the paper reports significant improvements in fidelity-oriented metrics like FID, Precision, and Density. However, these gains come at the cost of a severe reduction in Recall.

**Strengths:**

Improving the generation quality and fidelity of diffusion models is a central and important challenge in the field. The paper's goal of better capturing complex data structures is well-motivated. The proposed modification is a simple, single-line addition to the loss function.

**Weaknesses:**

1. Questionable Motivation.
The core argument of the paper appears to be based on a misunderstanding of the DDPM objective. The authors claim that the standard L2 loss $||\epsilon - \epsilon_\theta||^2$ does not explicitly impose structural (i.e., isotropic) information on the predicted noise. This is incorrect. The target noise $\epsilon$ is sampled from an isotropic Gaussian $\mathcal{N}(0, \mathbf{I})$. Minimizing the L2 distance between \epsilon_\theta and \epsilon is the most direct way to force \epsilon_\theta to have the same statistical properties as $\epsilon$, including isotropy.
The proposed regularizer, $\lambda(E[||\epsilon_\theta||^2/n] - 1)^2$, explicitly penalizes the second moment of the predicted noise if it deviates from that of a standard normal distribution. The standard L2 objective already does this implicitly, and more strongly, by matching the entire distribution, not just its second moment. Therefore, the regularizer appears redundant. The paper fails to provide a convincing explanation for why the standard objective fails in a way that this specific regularizer can fix. The highly abstract "metric space" argument in Section 3.3 obfuscates rather than clarifies this point, lacking a concrete connection to the actual training dynamics.

2. Unanalyzed and Severe Fidelity-Diversity Trade-off.
The experimental results show a clear and drastic pattern: I-Diff significantly improves Precision and Density while catastrophically reducing Recall (e.g., -69.8% on Oxford Flower, -58.4% on Pets). This is a classic fidelity-diversity trade-off, where the model learns to generate samples only from the densest parts of the true distribution (mode-seeking behavior), forfeiting its ability to cover the full diversity of the data.
While the authors acknowledge this by saying the method "is capable of capturing highly dense mode regions" (line 413) they fail to analyze it as a fundamental trade-off. For instance, by plotting Precision-Recall curves for different values of the regularization parameter $\lambda$ and comparing them to other techniques that navigate this trade-off (e.g., guidance scale, truncation). As it stands, the paper presents a method that severely damages diversity to improve fidelity, without providing the analysis needed to understand or control this compromise.

3. Inconsistent Gains and Potentially Weak Baselines.
The reported 47% FID reduction on CIFAR-100 for the DDPM baseline is a bold claim for such a simple regularizer, consider the reported 3.17 FID in original DDPM paper. However, the improvement on the LDM baseline for the same dataset is a much more modest 6.4%. This large discrepancy suggest that the vanilla DDPM baseline may be poorly tuned or suboptimal, and the regularizer is acting as a stabilizer for an otherwise brittle training process, rather than providing the claimed "structural information".

**Questions:**

1. Could you please provide a clear, empirical, or theoretical justification for why the standard $||\epsilon - \epsilon_\theta||^2$ objective is insufficient for making the predicted noise $\epsilon_\theta$ isotropic? Given that the L2 loss directly minimizes the distance to an isotropic target, what specific failure mode does your regularizer address that the L2 loss does not? An analysis of the moments of $\epsilon_\theta$ during training with and without the regularizer would be insightful.

2. The dramatic drop in Recall suggests your method induces strong mode-seeking behavior. Could you provide a more thorough analysis of this trade-off? For example, by showing Precision-Recall curves as $\lambda$ is varied, and comparing these curves to those produced by adjusting classifier-free guidance scale, which is a standard way to control this trade-off. Is there an operating point where fidelity is improved without such a catastrophic loss of diversity?

3. The difference in improvement between the DDPM (47% FID reduction on CIFAR-100) and LDM (6.4% FID reduction) is stark. Could this indicate that the baseline DDPM implementation was not fully optimized? Can you provide evidence that the regularizer's benefit is not simply stabilizing a weak baseline?

4. The motivation in Section 3.3 uses a highly abstract "metric space of distributions." Could you provide a more concrete, intuitive explanation of how the regularizer works? For example, how does forcing the magnitude of the predicted noise vector to be correct (which is what your regularizer does) help the model learn "complex geometries" in a way that matching the full noise vector does not?

---

### Official Review · Reviewer_jTT1 · 2025-10-31

**Soundness:** 2
**Presentation:** 2
**Contribution:** 2
**Rating:** 4
**Confidence:** 3

**Summary:**

This paper points out the limitations of DDPMs, which solely rely on the L2 norm for loss calculation.
It proposes a regularizer to enforce and strengthen the isotropy of the epsilon predicted by the DDPM.
By adding this regularizer to the simplified epsilon loss during training, the paper demonstrates clear improvements in FID, precision, and density.
This suggests that the proposed method enables a more accurate and faithful modeling of the true data distribution.
The method shows effectiveness on small-scale 2D data and various image datasets.

**Strengths:**

- The method is simple to implement, and its motivation is strong.

- It is a method that could potentially be applied to other types of diffusion models, not just DDPM.

**Weaknesses:**

- The effectiveness is primarily demonstrated on the somewhat out-dated DDPM framework. It would be more compelling if it also showed positive results on newer and simpler models like flow matching [A] or EDM [B].

- Performance improvements are only shown for unconditional generation. It remains questionable whether this method would be effective in conditional generation that uses classifier-free guidance (CFG) [C]. Since CFG also improves precision at the cost of diversity, it is unclear if this method would have a synergistic effect with CFG or if it would simply be redundant.

[A] Lipman et al., Flow Matching for Generative Modeling, ICLR 2023.

[B] Karras et al., Elucidating the Design Space of Diffusion-Based Generative Models, NeurIPS 2022.

[C] Ho et al., Classifier-Free Diffusion Guidance

**Questions:**

An experiment in the paper mentions "LDM" using only the acronym. The reviewer is curious if this stands for "Latent Diffusion Model".

---

### Official Review · Reviewer_d1au · 2025-10-31

**Soundness:** 3
**Presentation:** 2
**Contribution:** 2
**Rating:** 4
**Confidence:** 4

**Summary:**

This paper proposes I-Diff, a simple, model-agnostic regularizer that augments the standard DDPM/LDM noise-prediction loss with an isotropy term so the reverse process prefers solutions that not only match L2 error but also respect distributional structure.Intuitively, when multiple reverse paths are L2-indistinguishable, the isotropy term separates them by structure (Sec. 3.3/Fig. 2), guiding sampling toward high-density regions. Experiments on 2D toy data and small image datasets (CIFAR-10/100, Oxford Flowers/Pets) show consistent gains in Precision/Density and lower FID in several cases for both pixel-space DDPM and Latent Diffusion, with a modest Recall trade-off (Sec. 6; Tables 1–3).

**Strengths:**

- The method is a one-line change to the loss, architecture-agnostic, and accompanied by a clean metric-space justification that explains why it can help when L2 alone is ambiguous (Sec. 3.3; Appendix A.1).
- The paper explicitly argues that PRDC complements FID/IS and interprets the observed precision–recall trade-off rather than hiding it (Sec. 6.2, discussion under Table 3), which strengthens the causal link between the isotropy prior and improved structural alignment

**Weaknesses:**

The experimental scope is limited: results are confined to small-scale image datasets and 2D synthetics. There is no ImageNet-scale evaluation and no text-to-image benchmarks, so scalability and generality to modern T2I settings remain untested (Appendix A.3 lists only CIFAR-10/100, Oxford Flowers/Pets; A.4 “Limitations” positions the work as a foundational proof-of-concept and calls for broader datasets). Moreover, while PRDC is informative, the paper leans on it heavily; some tables show Recall/Coverage drops, indicating a tighter focus on high-density modes that may reduce diversity (Sec. 6/Table 3 commentary). Together, these issues make the case less conclusive for large-scale or conditional generative tasks, and future work should verify the approach on ImageNet and text-conditional diffusion models.

**Questions:**

See weaknesses.

---

### Note · Authors · 2025-11-15

I have read and agree with the venue's withdrawal policy on behalf of myself and my co-authors.